# Non-linear integral equations for the XXX spin-1/2 quantum chain with non-diagonal boundary fields

Holger Frahm[1], Andreas Klümper[2], Dennis Wagner[2], and Xin Zhang[3]

[1]Institut für Theoretische Physik, Leibniz Universität Hannover, Appelstr. 2, 30167 Hannover, Germany
[2]Department of Physics, University of Wuppertal, Gaussstraße 20, 42119 Wuppertal, Germany
[3]Beijing National Laboratory for Condensed Matter Physics, Institute of Physics, Chinese Academy of Sciences, Beijing 100190, China

November 25, 2025

## Abstract

The XXX spin-$\frac{1}{2}$ Heisenberg chain with non-diagonal boundary fields represents a cornerstone model in the study of integrable systems with open boundaries. Despite its significance, solving this model exactly has remained a formidable challenge due to the breaking of $U(1)$ symmetry. Building on the off-diagonal Bethe Ansatz (ODBA), we derive a set of nonlinear integral equations (NLIEs) that encapsulate the exact spectrum of the model.

For $U(1)$ symmetric spin-$\frac{1}{2}$ chains such NLIEs involve two functions $a(x)$ and $\bar{a}(x)$ coupled by an integration kernel with short-ranged elements. The solution functions show characteristic features for arguments at some length scale which grows logarithmically with system size $N$.

In the case considered here the $U(1)$ symmetry is broken by the non-diagonal boundary fields and the equations involve a novel third function $c(x)$, which captures the inhomogeneous contributions to the $T$-$Q$ equation in the ODBA. The kernel elements coupling this function to the standard ones are long-ranged and lead for the ground-state to a winding phenomenon. In $\log(1 + a(x))$ and $\log(1 + \bar{a}(x))$ we observe a steep change by $2\pi\mathrm{i}$ at a characteristic scale $x_1$ of the argument. Other features appear at a value $x_0$ which is of order $\log N$. These two length scales, $x_1$ and $x_0$, are independent: their ratio $x_1/x_0$ is large for small $N$ and small for large $N$. Explicit solutions to the NLIEs are obtained numerically for these limiting cases, though intermediate cases ($x_1/x_0 \sim 1$) present computational challenges.

This work lays the foundation for studying finite-size corrections and conformal properties of other integrable spin chains with non-diagonal boundaries, opening new avenues for exploring boundary effects in quantum integrable systems.

## 1 Introduction

The XXX spin-$\frac{1}{2}$ chain with periodic boundary conditions is a seminal representative of quantum integrable systems. A mathematically sufficient condition for integrability is the Yang-Baxter equation [1], a fundamental tool that has shaped much of our modern understanding and has led to the dicovery of many new integrable systems. While the XXX chain with periodic boundary conditions was solved by Bethe in the famous paper [2], integrability in the presence of boundaries introduces rich and intricate challenges.

Non periodic boundary conditions significantly influence integrable systems, and their exploration has a long history. For instance, the single-component Bose gas with delta-function interactions and open boundaries was solved in [3]. The XXX spin-$\frac{1}{2}$ chain with parallel boundary fields has been successfully solved using both coordinate and algebraic Bethe Ansatz methods [4, 5], with finite-size corrections studied in [6].

Extending these goals to more general boundary conditions and models required new theoretical tools, such as Sklyanin's reflection algebra [5, 7], which elegantly accounts for the factorization of scattering processes at the chain

ends. This framework is the basis of the proof of integrability for the Heisenberg spin chain with general non-parallel boundary fields in [8].

However, non-parallel boundary fields break the $U(1)$ symmetry of the system, presenting a formidable challenge to traditional Bethe Ansatz methods. To address this, several advanced techniques have been developed. For example, T-Q relations have been applied to specific cases of the partially anisotropic Heisenberg model, the XXZ chain, such as for root of unity cases of the anisotropy [9] or for special choices of the boundary parameters [10, 11]. Fusion techniques have also provided insights through hierarchies of transfer matrices satisfying $T$- and $Y$- systems [12]. More recently, the off-diagonal Bethe Ansatz (ODBA) [13, 14] introduced an elegant framework leveraging commuting transfer matrices and inhomogeneous T-Q relations. Two such formulations of the ODBA for the XXX spin-$\frac{1}{2}$ chain have been developed [14], and the completeness of one of these approaches [13] has been argued in [15].

Alternative methods have also emerged, including the modified algebraic Bethe Ansatz, which incorporates chiral basis states into the algebraic Bethe Ansatz framework [16–20], the chiral coordinate Bethe ansatz [21], and separation of variables techniques, which have been applied to increasingly general boundary conditions from [22, 23] to [24].

Recent studies have focused on deriving physical properties that are independent of the boundary field angles, such as the ground-state energy, surface energy, and low-lying excitations in the thermodynamic limit [25]. In this paper, we present a new analysis to the isotropic Heisenberg chain with arbitrary boundary fields, explicitly retaining terms that depend on the angle between the boundary fields. Starting from the inhomogeneous T-Q relation with two $Q$-functions [13, 14], we derive the ground-state energy, surface energy, and finite-size corrections, offering a comprehensive perspective on these fundamental quantities.

The paper is organized as follows. In Section 2 we summarize the results of the ODBA approach [13, 14] which we use as our starting point. In this paper we focus on the case of negative $p, q$ parameters (negative longitudinal components of the boundary fields) and leave other combinations for a future publication. In Section 3 we identify useful auxiliary functions that satisfy a set of functional equations. These equations are rewritten as linear equations for the Fourier transforms of the auxiliary functions. By use of the analyticity of the eigenvalue function these linear equations close. Section 4 presents the analytic results for the bulk and surface energies and numerical results for the auxiliary functions. The paper closes with conclusions in Section 6.

## 2 The inhomogeneous T-Q relation

For the spin-1/2 Heisenberg model with isotropic bulk interaction the Hamiltonian of the system with arbitrary boundary fields can be brought to the form

$$H = \sum_{j=1}^{N-1} \vec{\sigma}_j \cdot \vec{\sigma}_{j+1} + \frac{1}{p}\sigma_1^z + \frac{1}{q}\left(\sigma_N^z + \xi\sigma_N^x\right),$$

where $N$ is the number of sites, and $p$, $q$, and $\xi$ are boundary parameters.

We start by using the inhomogeneous T-Q relation of [13, 14] with two $Q$-functions. We introduce new combinations of the boundary parameters that are useful for our purposes

$$\xi_1 := (1 + \xi^2)^{1/2}, \qquad p_1 := -2p, \qquad p_2 := -2q/\xi_1. \tag{1}$$

We study the eigenvalue function for the associated transfer matrix for an even number of sites $N$ and rescale the function by dropping a constant factor $(\mathrm{i}/2)^{2N+2}$

$$\Lambda(x) = \lambda_1(x) + \lambda_2(x) + \lambda_3(x), \tag{2}$$

where we also reparameterized the argument $u = \mathrm{i}x/2 - 1/2$ used in [14] by the new variable $x$. The summands in (2) are

$$\lambda_1(x) := \phi_1(x)\frac{q_1(x+2\mathrm{i})}{q_2(x)}, \quad \lambda_2(x) := \frac{\phi_2(x)}{q_1(x)q_2(x)}, \quad \lambda_3(x) := \phi_3(x)\frac{q_2(x-2\mathrm{i})}{q_1(x)}, \tag{3}$$

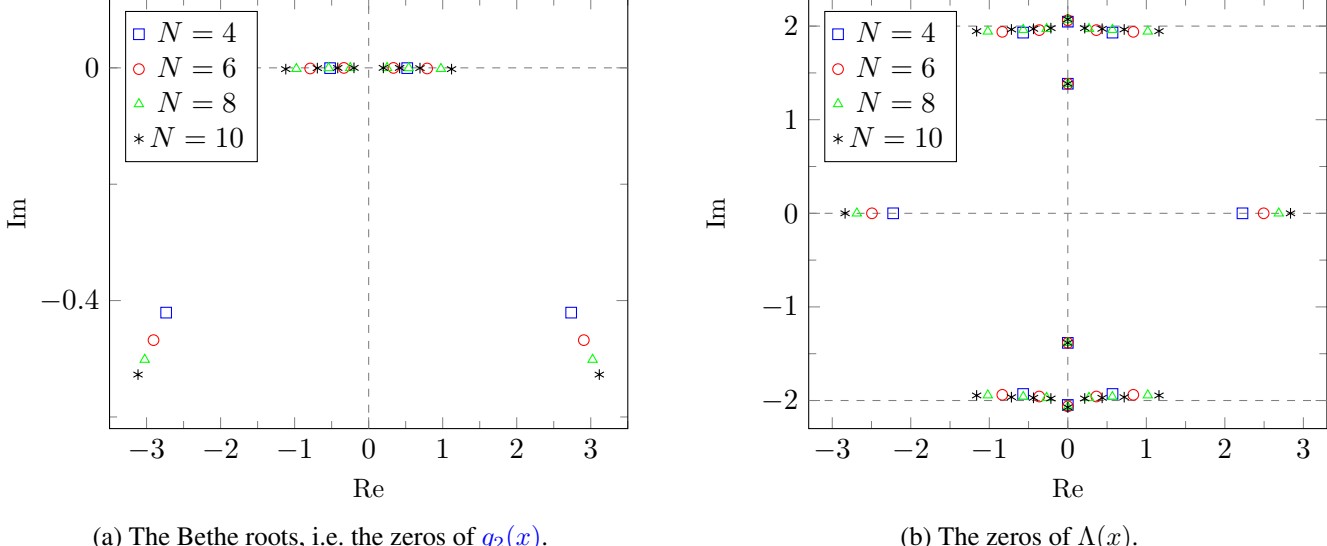

(a) The Bethe roots, i.e. the zeros of $q_2(x)$.

(b) The zeros of $\Lambda(x)$.

Figure 1: Depiction of Bethe roots and zeros of the eigenvalue function for the ground-state and parameters $\{p, q, \xi\} = \{-0.6, -0.3, 1.2\}$. This choice is representative of the situation in which both $p$ and $q$ are negative. Note the existence of two extremal roots with large negative imaginary parts. All other roots lie very close to the real axis.

where $q_1(x)$ and $q_2(x)$ are polynomials of degree $N$ with $q_1(x) = q_2(-x)$. The zeros of $q_2(x)$ are called Bethe roots. See Fig.1a for the depiction of zeros of $q_2(x)$ for short system size.

The functions $\phi_1(x)$, $\phi_2(x)$, $\phi_3(x)$ are explicitly given by

$$
\begin{aligned}
\phi_1(x) &:= \xi_1 \frac{\varphi(x)}{x}(x - \mathrm{i})^{2N+1}, &\qquad \varphi(x) &:= (x + \mathrm{i} + \mathrm{i}p_1)(x + \mathrm{i} + \mathrm{i}p_2), \\
\phi_2(x) &:= 2\big(1 - \xi_1\big)(x^2 + 1)^{2N+1}, \\
\phi_3(x) &:= \xi_1 \frac{\overline{\varphi}(x)}{x}(x + \mathrm{i})^{2N+1}, &\qquad \overline{\varphi}(x) &:= (x - \mathrm{i} - \mathrm{i}p_1)(x - \mathrm{i} - \mathrm{i}p_2).
\end{aligned} \tag{4}
$$

In [14] the Bethe roots are calculated for relatively short chains by numerically solving for the Bethe equations. These equations are derived from the condition that all potential poles in (2) cancel resulting in an analytic function $\Lambda(x)$. We do not write down these equations explicitly as we do not use them.

## 3 Transformation of functional equations to integral equations

Here we adopt a different resp. opposite approach to deriving Bethe ansatz equations from the analyticity of the eigenvalue function. We avoid the calculation of the Bethe roots and use the analyticity of $\Lambda(x)$ itself. The entire reasoning is based on identifying a set of suitable analytic functions satisfying sufficiently many functional equations. These will be rewritten by use of the Fourier transform into (non-linear) integral equations of convolution type. In analogy to [26, 27], we find that the following functions are useful

$$
b(x) := \frac{\lambda_2(x) + \lambda_3(x)}{\lambda_1(x)}, \qquad\qquad B(x) := 1 + b(x) = \frac{\Lambda(x)}{\lambda_1(x)}, \tag{5}
$$

$$
\overline{b}(x) := \frac{\lambda_1(x) + \lambda_2(x)}{\lambda_3(x)}, \qquad\qquad \overline{B}(x) := 1 + \overline{b}(x) = \frac{\Lambda(x)}{\lambda_3(x)}, \tag{6}
$$

$$
c(x) := \frac{\lambda_2(x)\Lambda(x)}{\lambda_1(x)\lambda_3(x)}, \qquad\qquad C(x) := 1 + c(x) = \frac{[\lambda_1(x) + \lambda_2(x)][\lambda_2(x) + \lambda_3(x)]}{\lambda_1(x)\lambda_3(x)}. \tag{7}
$$

Note that $b$, $\bar{b}$, $c$ are defined in terms of the (not explicitly known) functions appearing in (2). These functions plus the constant 1 are called $B$, $\overline{B}$, $C$ and also factorize into the same factors $\lambda_i$, $\lambda_1 + \lambda_2$, $\lambda_2 + \lambda_3$ as well as $\Lambda$.

The asymptotic behaviour of the functions for large arguments is

$$\Lambda(x) \simeq 2 \cdot x^{2N+2},$$
$$b(\infty) = \bar{b}(\infty) = 2/\xi_1 - 1, \qquad c(\infty) = 4(1/\xi_1^2 - 1/\xi_1). \tag{8}$$

The function $\Lambda(x)$ is analytic, even and possesses a number $2N + 2$ of zeros.

Numerical calculations for the largest eigenvalue of the transfer matrix–corresponding to the lowest eigenvalue of the Hamiltonian–show that $N$ zeros of $\Lambda(x)$ have imaginary parts greater than $+1$ with most of them close to $+2$, the same number of zeros have imaginary parts less than $-1$ and most of them close to $-2$, and two zeros $\pm x_0$ lie on the real axis. See Fig. 1b for the depiction of zeros of $\Lambda(x)$ for short system size.

By use of the above definitions and of the cancellation of poles in (2) resp. the Bethe ansatz equations we conclude that

$$D(x) := \frac{x q_2(x)[\lambda_2(x) + \lambda_3(x)]}{(x + \mathrm{i})^{2N+1}}, \quad \overline{D}(x) := \frac{x q_1(x)[\lambda_1(x) + \lambda_2(x)]}{(x - \mathrm{i})^{2N+1}}, \tag{9}$$

are polynomials of degree $N + 2$.

The rational functions $b$, $\bar{b}$, $c$, $B$, $\overline{B}$, $C$ allow for the following factorizations in terms of polynomials

$$b(x) = \text{cst.} \frac{(x + \mathrm{i})^{2N+1}}{\varphi(x)(x - \mathrm{i})^{2N+1}} \frac{D(x)}{q_1(x + 2\mathrm{i})},$$
$$\bar{b}(x) = \text{cst.} \frac{(x - \mathrm{i})^{2N+1}}{\overline{\varphi}(x)(x + \mathrm{i})^{2N+1}} \frac{\overline{D}(x)}{q_2(x - 2\mathrm{i})},$$
$$c(x) = \text{cst.} \frac{x^2}{\varphi(x)\overline{\varphi}(x)} \frac{\Lambda(x)}{q_1(x + 2\mathrm{i})q_2(x - 2\mathrm{i})},$$
$$B(x) = \text{cst.} \frac{x}{\varphi(x)(x - \mathrm{i})^{2N+1}} \frac{q_2(x)}{q_1(x + 2\mathrm{i})} \Lambda(x),$$
$$\overline{B}(x) = \text{cst.} \frac{x}{\overline{\varphi}(x)(x + \mathrm{i})^{2N+1}} \frac{q_1(x)}{q_2(x - 2\mathrm{i})} \Lambda(x),$$
$$C(x) = \text{cst.} \frac{1}{\varphi(x)\overline{\varphi}(x)} \frac{D(x)\overline{D}(x)}{q_1(x + 2\mathrm{i})q_2(x - 2\mathrm{i})}, \tag{10}$$

where in principle we may identify the (non-zero) constants, but we will not need those.

We want to "solve" these functional equations of multiplicative type with constant shifts in the arguments. The first step is the application of the logarithm, turning the product form into additive form. In the second step we apply the Fourier transform which turns functions with shifts in the arguments to transforms with simple factors leading eventually to a set of linear equations. Care has to be taken that the Fourier transform exists and the region of convergence of the inverse transform, the Fourier representation, is wide enough for our purposes. For these reasons we deal with the logarithmic derivatives of the equations above upon which the polynomial factors turn into Fourier transformable functions. At the end of these procedures, the non-linear integral equations emerge. The strategies for solving these equations are largely numerical and are described in Sect. 4.

In case of the largest eigenvalue, $\Lambda(x)$ has two zeros in the strip $|\mathrm{Im}(x)| < 1 + \epsilon$ (where $\epsilon$ is small and positive) lying symmetrically at points $\pm x_0$ on the real axis. The function $c(x)$ has four zeros on the real axis, namely $\pm x_0$ and the zero at 0 of second order. Hence the logarithmic derivatives of $\Lambda(x)$ and $c(x)$ do not have Fourier representations in the strip $|\mathrm{Im}(x)| < 1 + \epsilon$. We therefore introduce the analytic and non-zero functions $\widetilde{\Lambda}(x)$, $\widetilde{c}(x)$ by

$$\Lambda(x) = \lambda_0(x)\widetilde{\Lambda}(x), \qquad\qquad\qquad \lambda_0(x) := (x - x_0)(x + x_0),$$
$$c(x) = x^2 \lambda_0(x)\widetilde{c}(x). \tag{11}$$

The logarithmic derivatives of $\widetilde{\Lambda}(x)$ and $\widetilde{c}(x)$ have Fourier representations in suffiently wide strips around the real axis.

We use the Fourier transform pair

$$\widetilde{f}(k) = \frac{1}{2\pi} \int_{-\infty}^{\infty} dx\, e^{-ikx} f(x), \qquad f(x) = \int_{-\infty}^{\infty} dk\, e^{ikx} \widetilde{f}(k), \tag{12}$$

with the alternative notation $FT[f] \equiv \widetilde{f}$ where the standard argument of $FT[f]$ is an implicit $k$. Note that

$$f(x) \leftrightarrow \widetilde{f}(k) \quad \text{implies} \quad f(x + ic) \leftrightarrow e^{-ck} \widetilde{f}(k). \tag{13}$$

We will often use the simple explicit Fourier transforms of (the logarithmic derivatives of) linear factors

$$FT\left[\frac{d}{dx}\log(x+ic)\right] =_{\text{Im}(c)>0} \begin{cases} -ie^{-ck}, & k > 0, \\ 0, & k < 0, \end{cases} \tag{14}$$

$$=_{\text{Im}(c)<0} \begin{cases} 0, & k > 0, \\ ie^{-ck}, & k < 0. \end{cases} \tag{15}$$

Next we introduce a shorthand for the Fourier transform of the logarithmic derivative of a function $f(x)$, which is very useful but potentially misleading

$$f := FT\left[\frac{d}{dx}\log f(x)\right], \tag{16}$$

i.e. the same symbol $f$ without specifying the argument $k$ shall mean a new function (in general) different from $f(x)$. At first sight this looks confusing if not non-sensic. However in practical calculations, misunderstandings are almost excluded: the same symbols appearing above in multiplicative relations now appearing in additive relations have a different meaning. This notation keeps the symbols manageable as the use of several levels of tildes, bars or indices is avoided.

From the above (14,15) we immediately calculate the transforms for the logarithmic derivatives of $\varphi(x)$ resp. $\overline{\varphi}(x)$ (noting that $p, q < 0$, i.e. $p_1, p_2 > 0$) and by use of the notation (16)

$$\varphi = \begin{cases} -i(e^{-(1+p_1)k} + e^{-(1+p_2)k}), & k > 0, \\ 0, & k < 0, \end{cases} \qquad \overline{\varphi} = \begin{cases} 0, & k > 0, \\ i(e^{(1+p_1)k} + e^{(1+p_2)k}), & k < 0, \end{cases} \tag{17}$$

The functions $q_1(x)$, $q_2(x)$, have zeros close to the real axis. For the logarithmic derivatives of $q_1(x)$ resp. $q_2(x)$ we use Fourier representations in the upper resp. lower half plane with vanishing Fourier components for $k < 0$ resp. $k > 0$. For $p_1, p_2 > 0$ and small system size the functions $D(x)$ resp. $\overline{D}(x)$ have zeros in the upper resp. the lower half-plane. This is also the case for all system sizes if $p_1, p_2 > 1$ and certain cases of $p_1 > 1 > p_2 > 0$ which is what we consider in this paper. In any case, for large system size the bulk of the zeros of $D(x)$ resp. $\overline{D}(x)$ is close to $\text{Im}(x) = +2$ resp. $\text{Im}(x) = -2$. For the logarithmic derivatives of $D(x)$ resp. $\overline{D}(x)$ we use Fourier representations in the semi-planes $\text{Im}(x) < +2$ resp. $\text{Im}(x) > -2$ with vanishing Fourier components for $k > 0$ resp. $k < 0$.

Now we apply the described transform to (10) and obtain

$$\begin{aligned}
\text{for } k > 0: \quad & b = -(2N+1)ie^{-k} - \varphi - e^{-2k}q_1, \\
& \overline{b} = (2N+1)ie^{-k} + \overline{D}, \\
& \widetilde{c} = -\varphi - e^{-2k}q_1 + \widetilde{\Lambda}, \\
& B = -\varphi - e^{-2k}q_1 + \widetilde{\Lambda}, \\
& \overline{B} = -i + (2N+1)ie^{-k} - 2i\cos(kx_0) + q_1 + \widetilde{\Lambda}, \\
& C = -\varphi + \overline{D} - e^{-2k}q_1,
\end{aligned} \tag{18}$$

and

$$\text{for } k < 0: \qquad b = -(2N+1)\mathrm{i}\mathrm{e}^k + D,$$
$$\bar{b} = (2N+1)\mathrm{i}\mathrm{e}^k - \overline{\varphi} - \mathrm{e}^{2k}q_2,$$
$$\widetilde{c} = -\overline{\varphi} - \mathrm{e}^{2k}q_2 + \widetilde{\Lambda},$$
$$B = \mathrm{i} - (2N+1)\mathrm{i}\mathrm{e}^k + 2\mathrm{i}\cos(kx_0) + q_2 + \widetilde{\Lambda},$$
$$\overline{B} = -\overline{\varphi} - \mathrm{e}^{2k}q_2 + \widetilde{\Lambda},$$
$$C = -\overline{\varphi} + D - \mathrm{e}^{2k}q_2. \tag{19}$$

The last three equations of (18) can be solved for $q_1, \overline{D}, \widetilde{\Lambda}$ in terms of $B, \overline{B}, C$ and this inserted into the first three equations gives

$$\text{for } k > 0: \qquad b = \frac{-\mathrm{i} - (2N+1)\mathrm{i}\mathrm{e}^k - \mathrm{e}^{2k}\varphi - 2\mathrm{i}\cos(kx_0)}{1 + \mathrm{e}^{2k}} + \frac{B - \overline{B}}{1 + \mathrm{e}^{2k}},$$
$$\bar{b} = \frac{\mathrm{i} + (2N+1)\mathrm{i}\mathrm{e}^k + \mathrm{e}^{2k}\varphi + 2\mathrm{i}\cos(kx_0)}{1 + \mathrm{e}^{2k}} - \frac{B - \overline{B}}{1 + \mathrm{e}^{2k}} + C,$$
$$\widetilde{c} = B,$$
$$\widetilde{\Lambda} = \frac{-(2N+1)\mathrm{i} + \mathrm{i}\mathrm{e}^k + \mathrm{e}^{3k}\varphi + 2\mathrm{i}\mathrm{e}^k\cos(kx_0)}{\mathrm{e}^k(1 + \mathrm{e}^{2k})} + \frac{\mathrm{e}^{2k}B + \overline{B}}{1 + \mathrm{e}^{2k}}. \tag{20}$$

The last three equations of (19) can be solved for $q_2, D, \widetilde{\Lambda}$ in terms of $B, \overline{B}, C$ and this inserted into the first three equations gives

$$\text{for } k < 0: \qquad b = \frac{-\mathrm{i}\mathrm{e}^{2k} - (2N+1)\mathrm{i}\mathrm{e}^k + \overline{\varphi} - 2\mathrm{i}\mathrm{e}^{2k}\cos(kx_0)}{1 + \mathrm{e}^{2k}} + \mathrm{e}^{2k}\frac{B - \overline{B}}{1 + \mathrm{e}^{2k}} + C,$$
$$\bar{b} = \frac{\mathrm{i}\mathrm{e}^{2k} + (2N+1)\mathrm{i}\mathrm{e}^k - \overline{\varphi} + 2\mathrm{i}\mathrm{e}^{2k}\cos(kx_0)}{1 + \mathrm{e}^{2k}} - \mathrm{e}^{2k}\frac{B - \overline{B}}{1 + \mathrm{e}^{2k}},$$
$$\widetilde{c} = \overline{B},$$
$$\widetilde{\Lambda} = \frac{(2N+1)\mathrm{i}\mathrm{e}^{3k} - \mathrm{i}\mathrm{e}^{2k} + \overline{\varphi} - 2\mathrm{i}\mathrm{e}^{2k}\cos(kx_0)}{(1 + \mathrm{e}^{2k})} + \frac{\mathrm{e}^{2k}B + \overline{B}}{1 + \mathrm{e}^{2k}}. \tag{21}$$

For many applications it is useful to consider these functions on shifted contours

$$a(x) := b(x - \mathrm{i}), \quad A(x) := B(x - \mathrm{i}), \quad \overline{a}(x) := \overline{b}(x + \mathrm{i}), \quad \overline{A}(x) := \overline{B}(x + \mathrm{i}), \tag{22}$$

because $a(x)$ and $\overline{a}(x)$ have a high order zero at $x = 0$ rendering these functions (and later $\log A(x)$ and $\log \overline{A}(x)$) very small on large parts of the real axis. It will appear that in some of the formulas below, the shift $x \to x \pm \mathrm{i}$ should be understood as a $x \to x \pm (1 - \epsilon)\mathrm{i}$ with small positive $\epsilon$.

The Fourier transforms of $a, A, \overline{a}, \overline{A}$ and $b, B, \overline{b}, \overline{B}$ are related by factors $\mathrm{e}^{\pm k}$

$$a = \mathrm{e}^k b, \quad A = \mathrm{e}^k B, \quad \overline{a} = \mathrm{e}^{-k}\overline{b}, \quad \overline{A} = \mathrm{e}^{-k}\overline{B}. \tag{23}$$

For these function we obtain the equations

$$\text{for } k > 0 \qquad a = \frac{-\mathrm{i} - (2N+1)\mathrm{i}\mathrm{e}^k - \mathrm{e}^{2k}\varphi - 2\mathrm{i}\cos(kx_0)}{\mathrm{e}^k + \mathrm{e}^{-k}} + \frac{\mathrm{e}^{-k}}{\mathrm{e}^k + \mathrm{e}^{-k}}(A - \mathrm{e}^{2k}\overline{A}),$$
$$\overline{a} = \frac{\mathrm{i}\mathrm{e}^{-2k} + (2N+1)\mathrm{i}\mathrm{e}^{-k} + \varphi + 2\mathrm{i}\mathrm{e}^{-2k}\cos(kx_0)}{\mathrm{e}^k + \mathrm{e}^{-k}} - \frac{\mathrm{e}^{-k}}{\mathrm{e}^k + \mathrm{e}^{-k}}(\mathrm{e}^{-2k}A - \overline{A}) + \mathrm{e}^{-k}C,$$
$$\widetilde{c} = \mathrm{e}^{-k}A,$$
$$\widetilde{\Lambda} = \frac{-(2N+1)\mathrm{i}\mathrm{e}^{-2k} + \mathrm{i}\mathrm{e}^{-k} + \mathrm{e}^k\varphi + 2\mathrm{i}\mathrm{e}^{-k}\cos(kx_0)}{\mathrm{e}^k + \mathrm{e}^{-k}} + \frac{A + \overline{A}}{\mathrm{e}^k + \mathrm{e}^{-k}}, \tag{24}$$

and

$$\text{for } k < 0 \qquad a = \frac{-\mathrm{i}\mathrm{e}^{2k} - (2N+1)\mathrm{i}\mathrm{e}^k + \overline{\varphi} - 2\mathrm{i}\mathrm{e}^{2k}\cos(kx_0)}{\mathrm{e}^k + \mathrm{e}^{-k}} + \frac{\mathrm{e}^k}{\mathrm{e}^k + \mathrm{e}^{-k}}(A - \mathrm{e}^{2k}\overline{A}) + \mathrm{e}^k C,$$

$$\overline{a} = \frac{\mathrm{i} + (2N+1)\mathrm{i}\mathrm{e}^{-k} - \mathrm{e}^{-2k}\overline{\varphi} + 2\mathrm{i}\cos(kx_0)}{\mathrm{e}^k + \mathrm{e}^{-k}} - \frac{\mathrm{e}^k}{\mathrm{e}^k + \mathrm{e}^{-k}}(\mathrm{e}^{-2k}A - \overline{A}),$$

$$\widetilde{c} = \mathrm{e}^k \overline{A},$$

$$\widetilde{\Lambda} = \frac{(2N+1)\mathrm{i}\mathrm{e}^{2k} - \mathrm{i}\mathrm{e}^k + \mathrm{e}^{-k}\overline{\varphi} - 2\mathrm{i}\mathrm{e}^k\cos(kx_0)}{\mathrm{e}^k + \mathrm{e}^{-k}} + \frac{A + \overline{A}}{\mathrm{e}^k + \mathrm{e}^{-k}}. \tag{25}$$

Finally we have to carry out the inverse Fourier transform which yields us expressions for the derivatives of the functions $\log a(x)$, $\log \overline{a}(x)$, $\log c(x)$ as sums of explicit functions and convolution integrals of explicit functions with the derivatives of the functions $\log A(x)$, $\log \overline{A}(x)$, $\log C(x)$. After deriving these equations we take the integral and determine the constants of integration.

As a first step we carry out the inverse Fourier transforms of the explicit functions. These functions are related to the digamma function $\psi$. By use of the integral formula

$$\psi(z) = \int_0^\infty \left[ \mathrm{e}^{-t} - \frac{1}{(1+t)^z} \right] \frac{dt}{t}, \tag{26}$$

and the substitution $1 + t = \mathrm{e}^x$ we find

$$\psi(z + 1/2) - \psi(z) = 4 \int_0^\infty dk \, \frac{\mathrm{e}^{-(4z-1)k}}{\mathrm{e}^k + \mathrm{e}^{-k}}. \tag{27}$$

This yields for a typical combination of terms appearing in the equation for $\log \Lambda(x)$

$$\ell(x, r) := \int_{-\infty}^0 dk \, \frac{\mathrm{i}\,\mathrm{e}^{rk}}{\mathrm{e}^k + \mathrm{e}^{-k}} \mathrm{e}^{\mathrm{i}kx} - \int_0^\infty dk \, \frac{\mathrm{i}\,\mathrm{e}^{-rk}}{\mathrm{e}^k + \mathrm{e}^{-k}} \mathrm{e}^{\mathrm{i}kx}$$

$$= \frac{\mathrm{i}}{4} \left[ \psi\left(\tfrac{1}{4}(r + 3 + \mathrm{i}x)\right) + \psi\left(\tfrac{1}{4}(r + 1 - \mathrm{i}x)\right) - \psi\left(\tfrac{1}{4}(r + 1 + \mathrm{i}x)\right) - \psi\left(\tfrac{1}{4}(r + 3 - \mathrm{i}x)\right) \right]. \tag{28}$$

Integrating this with respect to $x$ and suitably fixing the integration constant yields

$$L(x, r) := \log \frac{\Gamma\left(\tfrac{1}{4}(r + 3 + \mathrm{i}x)\right) \Gamma\left(\tfrac{1}{4}(r + 3 - \mathrm{i}x)\right)}{\Gamma\left(\tfrac{1}{4}(r + 1 + \mathrm{i}x)\right) \Gamma\left(\tfrac{1}{4}(r + 1 - \mathrm{i}x)\right)} + \log(4) = \log(x) + O\left(\frac{1}{x^2}\right), \tag{29}$$

where the asymptotics is given for large arguments.

The next combination appears in the expressions for $\log a(x)$ and $\log \overline{a}(x)$

$$\kappa(x, r) := \int_{-\infty}^0 dk \, \frac{\mathrm{e}^{rk}}{\mathrm{e}^k + \mathrm{e}^{-k}} \mathrm{e}^{\mathrm{i}kx} + \int_0^\infty dk \, \frac{\mathrm{e}^{-rk}}{\mathrm{e}^k + \mathrm{e}^{-k}} \mathrm{e}^{\mathrm{i}kx}$$

$$= \frac{1}{4} \left[ \psi\left(\tfrac{1}{4}(r + 3 + \mathrm{i}x)\right) + \psi\left(\tfrac{1}{4}(r + 3 - \mathrm{i}x)\right) - \psi\left(\tfrac{1}{4}(r + 1 + \mathrm{i}x)\right) - \psi\left(\tfrac{1}{4}(r + 1 - \mathrm{i}x)\right) \right]. \tag{30}$$

The integral of this function with respect to $x$ is called $\alpha(x, r)$

$$\alpha(x, r) := \mathrm{i} \log \frac{\Gamma\left(\tfrac{1}{4}(r + 3 - \mathrm{i}x)\right) \Gamma\left(\tfrac{1}{4}(r + 1 + \mathrm{i}x)\right)}{\Gamma\left(\tfrac{1}{4}(r + 3 + \mathrm{i}x)\right) \Gamma\left(\tfrac{1}{4}(r + 1 - \mathrm{i}x)\right)} \qquad \left( \to \pm \frac{\pi}{2} + \mathcal{O}\left(\frac{1}{x}\right) \text{ for } x \to \pm\infty \right). \tag{31}$$

Note that the introduced functions $\ell(x, r)$, $L(x, r)$, $\kappa(x, r)$, $\alpha(x, r)$ are real valued for real arguments $x$, $r$.

By use of these explicit functions we find for the eigenvalue function $\Lambda(x)$

$$\begin{aligned}
\log \Lambda(x) &= \log \widetilde{\Lambda}(x) + \log \lambda_0(x) \\
&= (2N+1)L(x, 2) - L(x, 1) + L(x, p_1) + L(x, p_2) \\
&\quad + \log(x^2 - x_0^2) - L(x - x_0, 1) - L(x + x_0, 1) \\
&\quad + e * (\log A + \log \overline{A}),
\end{aligned} \tag{32}$$

where the convolution $*$ of two functions $f, g$ is defined by

$$f * g(x) = \frac{1}{2\pi} \int dy\, f(x-y)g(y), \tag{33}$$

and the function $e(x)$ is given by the Fourier integral with explicit result

$$e(x) := \int_{-\infty}^{\infty} dk \frac{e^{ikx}}{e^k + e^{-k}} = \frac{\frac{\pi}{2}}{\cosh \frac{\pi}{2} x}. \tag{34}$$

The integral expressions for the functions $\log a(x)$, $\log \overline{a}(x)$, $\log c(x)$ are

$$\begin{aligned}
\log a(x) =& (2N+1) \log \tanh\left(\frac{\pi}{4} x\right) + \frac{\pi}{2} i + \log a_\infty - i\alpha(x - i, 1) \\
& + i\alpha(x - i, p_1) + i\alpha(x - i, p_2) - i\alpha(x - x_0 - i, 1) - i\alpha(x + x_0 - i, 1) \\
& + K_{11} * \log(A/A_\infty) + K_{12} * \log(\overline{A}/\overline{A}_\infty) + K_{13} * \log(C/C_\infty), \tag{35}
\end{aligned}$$

$$\begin{aligned}
\log \overline{a}(x) =& (2N+1) \log \tanh\left(\frac{\pi}{4} x\right) - \frac{\pi}{2} i + \log \overline{a}_\infty + i\alpha(x + i, 1) \\
& - i\alpha(x + i, p_1) - i\alpha(x + i, p_2) + i\alpha(x - x_0 + i, 1) + i\alpha(x + x_0 + i, 1) \\
& + K_{21} * \log(A/A_\infty) + K_{22} * \log(\overline{A}/\overline{A}_\infty) + K_{23} * \log(C/C_\infty), \tag{36}
\end{aligned}$$

$$\log c(x) = \log\left[c_\infty \cdot x^2 \left(x^2 - x_0^2\right)\right] + K_{31} * \log A/A_\infty) + K_{32} * \log(\overline{A}/\overline{A}_\infty) - 4 \log R, \tag{37}$$

where the asymptotic values $A_\infty = 1 + a_\infty$, $\overline{A}_\infty = 1 + \overline{a}_\infty$ and $C_\infty = 1 + c_\infty$ are easily obtained from (8) and $a_\infty = b_\infty := b(\infty)$ etc. The convolutions in (37) are done with $\log A$ and $\log \overline{A}$ evaluated on symmetric intervals $[-R, +R]$ with $R \to \infty$. The kernel matrix is

$$K(x) = \begin{bmatrix} \kappa(x, 1) & -\kappa(x - (2 - \epsilon)i, 1) & -i/(x - i) \\ -\kappa(x + (2 - \epsilon)i, 1) & \kappa(x, 1) & i/(x + i) \\ i/(x + i) & -i/(x - i) & 0 \end{bmatrix}, \qquad (\epsilon \text{ small and positive}). \tag{38}$$

Note that the careful prescription for the convolution integrals is necessary because of the slow asymptotics of $K_{31}$ and $K_{32}$ and the curious property that the functions $\log A$ and $\log \overline{A}$ show non-trivial windings at some argument $x_1$, see Fig. 2 for a system with size $N = 10^3$, still to be considered small on grounds that become clearer shortly. The equations (35), (36), (37) are non-linear integral equations (NLIEs) for the functions $a, \overline{a}, c$, because $A = 1 + a$, $\overline{A} = 1 + \overline{a}$, $C = 1 + c$. Numerical solutions are obtained by iterative treatments and evaluations of the convolution integrals by the Fast Fourier Transform. The kernel elements involving the $\epsilon$-regularization parameter have, in their Fourier representation, a product structure consisting of two factors: one that decays exponentially as argument $k \to +\infty$ (resp. $k \to -\infty$) and approaches a finite limit as $k \to -\infty$ (resp. $k \to +\infty$), and another factor of the form $\exp(\epsilon k)$ (resp. $\exp(-\epsilon k)$). These kernel elements are multiplied by Fourier transforms of functions that themselves decay exponentially as $k \to \pm\infty$. Consequently, it is justified to set $\epsilon = 0$ in this context. In parallel, the zero $x_0$ of $\Lambda(x)$ is found from solving $a(x_0 + i) = -1$ by evaluating the right hand side of (35) off the real axis. Another problem which is more serious is the treatment of the convolutions of the $\pm i/(x \pm i)$ kernels with the $\log A/A_\infty$ and $\log \overline{A}/\overline{A}_\infty$ functions. This results in a function with $-4 \log |x|$ asymptotics, clearly not to be done by numerical calculations. In the next section we show how to introduce counter-terms that can be treated analytically with a remaining convolution integral that can be done numerically.

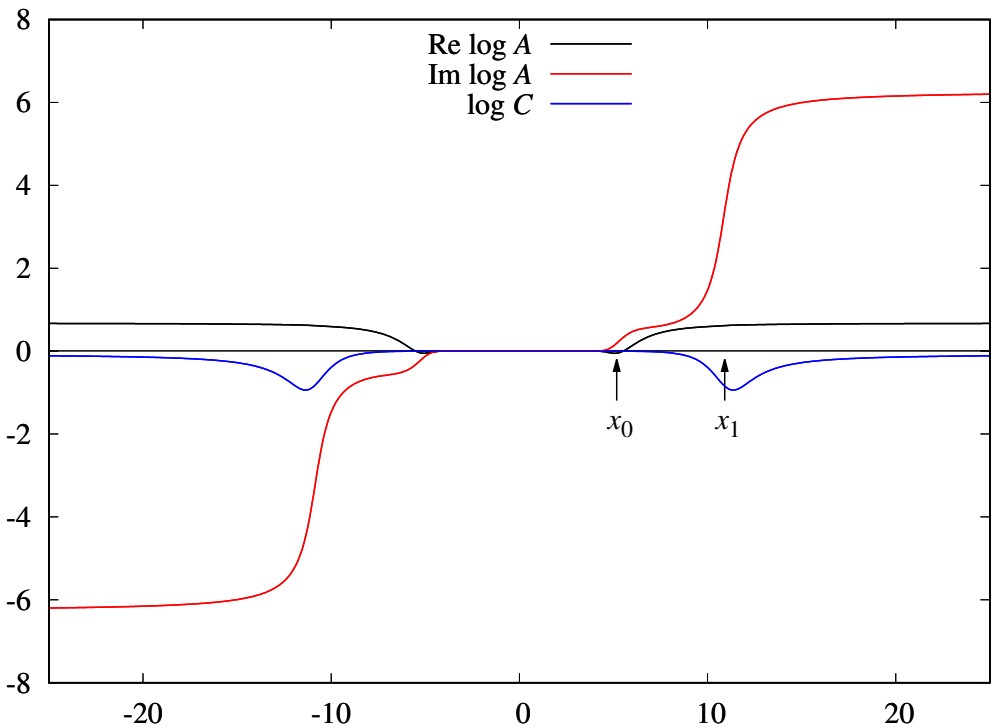

Figure 2: Graphs of the functions $\log A$ and $\log C$ for the ground-state eigenvalue of the spin-$1/2$ XXX chain with boundary parameters $p = -0.6$, $q = -0.3$, $\xi = 0.2$ and system size $N = 10^3$. The function $\log \overline{A}$ (not shown) is the complex conjugate of $\log A$. The function $\log C$ is real and even. The real (imaginary) part of $\log A$ is even (odd). There are two important length "scales" in the system. First of all, there are the positions $\pm x_0$ of the zeros of the function $\log C$ (remember $C = 1 + c$ and $c(\pm x_0) = 0$). Here $x_0 = 5.167.. \simeq \frac{2}{\pi} \log N$. In the plot, the zeros can not be read off, because the function $\log C$ looks flat in $[-x_0, +x_0]$ where it actually takes small positive values and noticeably negative values outside this interval. In the same interval $[-x_0, +x_0]$ the real and imaginary parts of $\log A$ take vanishingly small values, outside they approach non-zero asymptotics. The behaviour of the imaginary part of $\log A$ is most interesting. After developing into some shoulder for $x > x_0$ it takes a steady increase for arguments $x$ in the vicinity of some value $x_1 \sim 10.9$ ($> x_0$) and approaches $+2\pi$ for $x \to +\infty$. Note that in the vicinity of the same $\pm x_1$ the function $\log C$ shows minima.

## 4  Bulk and surface energies, first numerical results

There is a different and numerically better posed method to formulate the NLIEs than done above. We introduce simple subtraction terms in the convolution part such that the involved functions show vanishing asymptotics. And for the compensation we use counter terms in the driving (source) terms of the NLIEs. The proof of this form of NLIEs uses as intermediate step the differentiated form of the NLIEs where no issues of convergence arise. There the subtractions and compensation terms are introduced. They can be chosen to be simply of rational function type, i.e. the kernel $K$ convolved with suitable rational functions yields rational functions. After formulating the NLIEs for the derivatives $(\log a(x))'$,..., $(\log A(x))'$,... the NLIEs are integrated with respect to the argument $x$ and the integration constants are fixed by considering the limit $x \to \infty$. The result is

$$\begin{pmatrix} \log a \\ \log \overline{a} \\ \log c \end{pmatrix} = d + K * \begin{pmatrix} \log(A/A_\infty) - \log\left( \dfrac{x - x_{r+}}{x - x_{r-}} \cdot \dfrac{x - x_{l+}}{x - x_{l-}} \right) \\ \log(\overline{A}/\overline{A}_\infty) - \log\left( \dfrac{x - x_{r-}}{x - x_{r+}} \cdot \dfrac{x - x_{l-}}{x - x_{l+}} \right) \\ \log(C/C_\infty). \end{pmatrix}, \tag{39}$$

The parameters $x_{r+}$ and $x_{r-}$ are complex numbers with positive real part and positive resp. negative imaginary parts, $x_{l+}$ and $x_{l-}$ are defined similarly with negative real parts. The introduced functions subtract the winding behaviour observed in the functions $\log A$ and $\log \overline{A}$. Now the inhomogeneity $d$ is a tuple of three functions containing the counter terms

$$
\begin{aligned}
d_1(x) = {} & (2N+1)\log\tanh\left(\frac{\pi}{4}x\right) + \frac{\pi}{2}\mathrm{i} - \mathrm{i}\alpha(x-\mathrm{i},1) \\
& + \mathrm{i}\alpha(x-\mathrm{i},p_1) + \mathrm{i}\alpha(x-\mathrm{i},p_2) - \mathrm{i}\alpha(x-x_0-\mathrm{i},1) - \mathrm{i}\alpha(x+x_0-\mathrm{i},1) \\
& + \log\left(a_\infty \frac{x-x_{r+}-2\mathrm{i}}{x-x_{r-}}\cdot\frac{x-x_{l+}-2\mathrm{i}}{x-x_{l-}}\right),
\end{aligned}
\tag{40}
$$

$$
\begin{aligned}
d_2(x) = {} & (2N+1)\log\tanh\left(\frac{\pi}{4}x\right) - \frac{\pi}{2}\mathrm{i} + \mathrm{i}\alpha(x+\mathrm{i},1) \\
& - \mathrm{i}\alpha(x+\mathrm{i},p_1) - \mathrm{i}\alpha(x+\mathrm{i},p_2) + \mathrm{i}\alpha(x-x_0+\mathrm{i},1) + \mathrm{i}\alpha(x+x_0+\mathrm{i},1) \\
& + \log\left(\overline{a}_\infty \frac{x-x_{r-}+2\mathrm{i}}{x-x_{r+}}\cdot\frac{x-x_{l-}+2\mathrm{i}}{x-x_{l+}}\right),
\end{aligned}
\tag{41}
$$

$$
d_3(x) = \log\left(c_\infty \frac{x^2(x^2-x_0^2)}{(x-x_{r-}+\mathrm{i})(x-x_{r+}-\mathrm{i})(x-x_{l-}+\mathrm{i})(x-x_{l+}-\mathrm{i})}\right).
\tag{42}
$$

The concrete values of $x_{r\pm}$ and $x_{l\pm}$ drop out of the calculations. This is most transparently understood in the differentiated form of the expressions. The kernel elements $\kappa$ act in the combination of $\kappa(x,1)+\kappa(x\mp(2-\epsilon)\mathrm{i},1) = \pm[\mathrm{i}/(x\pm\mathrm{i}\epsilon)-\mathrm{i}/(x\mp 2\mathrm{i})]$ on a rational function, appearing from the derivatives of the introduced terms on the right hand side of (39), which gives a similar rational function with shifted zeros and poles. Likewise the action of the $\pm\mathrm{i}/(x\pm\mathrm{i})$ kernel elements yield related rational functions. These are the expressions that appear in (40)-(42). The chosen values of $x_{r\pm}$ and $x_{l\pm}$ affect at most the accuracy of the calculations. For practical purposes we choose for these numbers

$$
x_{r\pm} = \tilde{x}_1 \pm \mathrm{i}\,\tilde{\delta} \qquad x_{l\pm} = -\tilde{x}_1 \pm \mathrm{i}\,\tilde{\delta},
\tag{43}
$$

with some $\tilde{\delta} > 0$ and $\tilde{x}_1$ is an estimate of the location of the transition of the imaginary part of $\log A$ from small resp. practically zero values to $+2\pi$ as explained in Fig. 2.

A natural approach to the numerical iteration of the NLIEs is to choose initial data for $\log A$, $\log \overline{A}$, and $\log C$ in the form of functions whose qualitative behaviour resembles that of solutions obtained directly from the Bethe ansatz equations. Such calculations of course are possible only for small system sizes ($N \sim 10$). For such systems the function $a(x)$ takes vanishingly small values for arguments $x$ close to 0. Increasing $x$ beyond values of about $\frac{2}{\pi}\log N$ leads to a noticeable increase of the absolute value of $a(x)$ with some small positive angle with respect to the real axis. Then for larger values of the argument, at some $x_1$, the values of $a(x)$ move sharply into the complex plane around the point $-1$ which is encircled exactly once in counter-clockwise manner. This is the reason for the sharp increase of $\mathrm{Im}(\log A)$ in Fig. 2.

By use of qualitatively similar initial data for the iterative treatment of the NLIEs we found convergence for much larger system sizes like that shown in Fig. 2 for $N = 10^3$. This is achieved by working with for instance $2\log[\tanh(x-x_1-\mathrm{i}\pi/4)\tanh(x+x_1-\mathrm{i}\pi/4)]$ functions for the imaginary part of $\log A$ and estimates for $x_1$. The iterated application of the NLIEs on such initial data with a "good choice of the value" for $x_1$ leads to convergence: for values of $x_1$ chosen too large (small) the transition point of the resulting function moves to larger (smaller) values clearly marking an instability. However, increasing the system size further will ultimately and independently of the chosen boundary parameters lead to the loss of convergence.

Independent of the system size we found that the zero $x_0$ of the function $c(x)$ scales like $\frac{2}{\pi}\log N$. The point $x_0$ also separates ranges of the argument $x$ for which $a(x)$ takes vanishingly small values, due to the leading $\tanh^{2N+1}(\frac{\pi}{4}x)$ factor resulting from the NLIEs, and larger values of $x$ with $a(x)$ being of order 1. From the NLIEs it is also obvious and actually necessary that $a(x)$ describes the encircling of $-1$ and hence $\log A$ shows two times an increase of the imaginary part by $2\pi\mathrm{i}$. Then the function $\log c(x)$ has an asymptotic behaviour that is no longer of order $\log x^4$, but instead it is now the logarithm of a rational function with order 4 polynomials in the numerator and in the denominator. It is or seems absolutely natural that the encircling of $-1$ by $a(x)$ happens at arguments close to a value $x_1$ which is larger than $x_0$: only for $x > x_0$ the values $a(x)$ move away from 0.

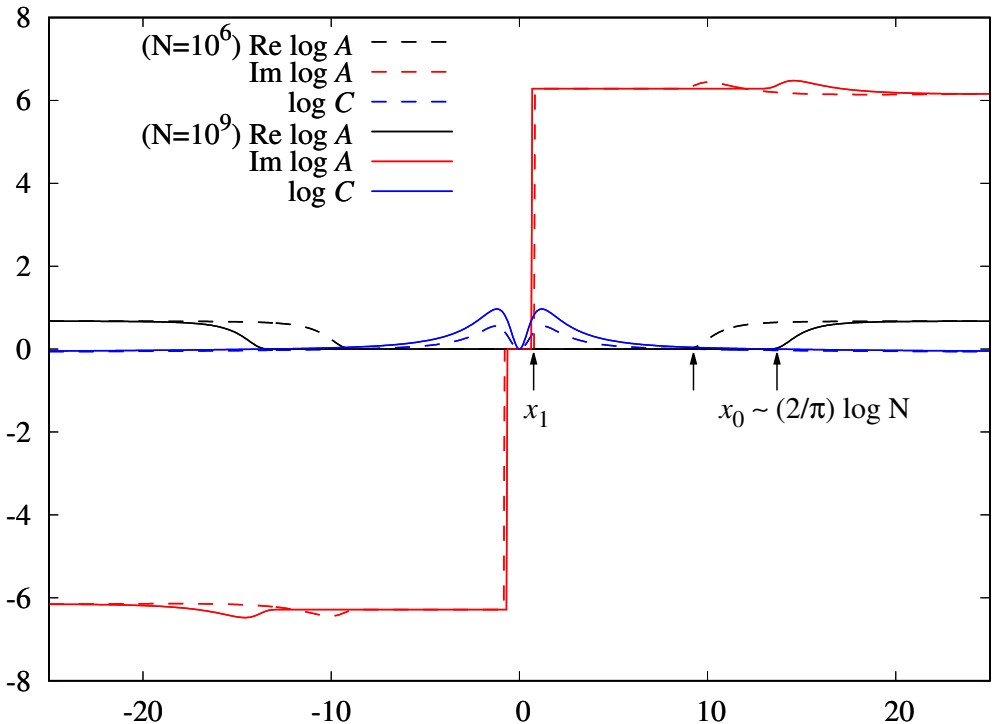

Figure 3: Plot of the functions $\log A$ and $\log C$ for the ground-state eigenvalue of the spin-$1/2$ XXX chain with parameters and colour coding as in Fig. 2 for larger system sizes $N = 10^6$ (dashed lines) $N = 10^9$ (solid lines). The function $\log C$ is zero at $\pm x_0$ with value $x_0 = 9.24...(13.68...) \simeq \frac{2}{\pi} \log N$ for $N = 10^6$ $(10^9)$. We note the qualitative differences to Fig. 2. Here, the point $x_1$, at which the imaginary part of $\log A$ increases sharply (!) from 0 to $+2\pi$, has a value less (!) than $x_0$. Within the resolution of the figure this is a step function. The point $x_1$ is rather well defined with value $x_1 = 0.83...(0.64...)$ for $N = 10^6$ $(10^9)$. Note that $x_0$ increases and $x_1$ decreases with $N$. The function $\log C$ takes positive and noticeably large values within the interval $[-x_0, +x_0]$ and rather flat negative values outside. In the neighbourhood of $\pm x_1$ of the sharp transitions of the imaginary part of $\log A$ the function $\log C$ shows maxima.

For large system sizes we simply did not find any suitable initial data set with the winding occurring at some $x_1 > x_0$ whether we chose $x_1$ much larger than $x_0$ or of similar size. For a failure of convergence many scenarios are conceivable, but the resolution of the problem appeared in the least expected manner. The winding happens in the "forbidden region" $[-x_0, +x_0]$ where the values of the function $a(x)$ seem to be tied to 0.

For sufficiently small system sizes we see that the zeros of $q_1(x)$ and $q_2(x)$ with largest positive (negative) real parts form complex conjugate pairs of a pole and a zero of the function $A(x)$. Following the function $A(x)$ for real values of $x$ straight through the complex conjugate pole/zero pair results into $A(x)$ performing a loop around 0. The scaling of this extremal pole/zero pair with increasing size $N$ is difficult to follow. Increasing $N$ from very small values shows a motion of the pairs away from the origin.

We postulate that for some system size this trend reverses and the extremal pairs move back to the origin from which they keep a finite distance in the limit $N \to \infty$. The condition for this to happen is that the separation of the zero and pole in the pair is approaching 0 exponentially fast. Under this condition, the above decribed winding happens so fast that the convolution integrals produce contributions that cancel the leading $(2N + 1) \log \tanh \frac{\pi}{4} x$. On the basis of this reasoning we found solutions of the NLIEs with much larger system sizes like those illustrated in Fig. 3.

Next we turn to the calculation of finite size-corrections. From the eigenvalue expression (32) the energy is

obtained by

$$
\begin{aligned}
E = -2\mathrm{i}\frac{\partial}{\partial x}\log\Lambda(x)\Big|_{-\mathrm{i}} - N = {}& -\,(2N+1)2\mathrm{i}\,\ell(-\mathrm{i},2) - N + 2\mathrm{i}\left[\ell(-\mathrm{i},1) - \ell(-\mathrm{i},p_1) - \ell(-\mathrm{i},p_2)\right] \\
& + 2\mathrm{i}\left[\frac{1}{\mathrm{i}+x_0} + \frac{1}{\mathrm{i}-x_0} + \ell(-\mathrm{i}-x_0,1) + \ell(-\mathrm{i}+x_0,1)\right] \\
& - 2\mathrm{i}\left[e' * (\log A + \log\overline{A})\big|_{-\mathrm{i}}\right].
\end{aligned}
\tag{44}
$$

By use of (28) we obtain

$$
2\mathrm{i}\,\ell(-\mathrm{i},r) = \psi\left(\frac{r+2}{4}\right) - \psi\left(\frac{r}{4}\right) - \frac{2}{r}.
\tag{45}
$$

Here and in the following calculations we make use of the functional equations and special argument identities

$$
\psi(x+1) = \psi(x) + \frac{1}{x},
$$
$$
\psi(1-x) - \psi(x) = \pi\cot(\pi x), \quad \psi(\tfrac{3}{4}) - \psi(\tfrac{1}{4}) = \pi,
$$
$$
\psi(1) - \psi(\tfrac{1}{2}) = 2\log 2.
$$

With these identities the first line of (44) gives the known bulk and boundary contributions to the ground state energy of the antiferromagnetic XXX spin-$1/2$ chain with boundary fields in the thermodynamic limit [28–30]:

$$
\begin{aligned}
E_0 ={}& N - 1 + \frac{2}{p_1} + \frac{2}{p_2} + N\epsilon_\infty + f_\infty + \mathcal{O}\left(\frac{1}{N}\right), \\
\epsilon_\infty ={}& -4\log 2, \quad f_\infty = \pi - 2\log 2 + \sum_{k=1,2}\left(\psi\left(\frac{p_k}{4}\right) - \psi\left(\frac{p_k+2}{4}\right)\right).
\end{aligned}
\tag{46}
$$

The surface energy also agrees with the XXX limits of the results in [31] and [32] (including $T \to 0$).

Similarly, one finds

$$
\begin{aligned}
2\mathrm{i}&\left[\frac{1}{\mathrm{i}+x_0} + \frac{1}{\mathrm{i}-x_0} + \ell(-\mathrm{i}-x_0,1) + \ell(-\mathrm{i}+x_0,1)\right] \\
&= \pi\cot\tfrac{\pi}{4}(1+\mathrm{i}x_0) + \pi\cot\tfrac{\pi}{4}(1-\mathrm{i}x_0) = \frac{2\pi}{\cosh\frac{\pi}{2}x_0} = 4e(x_0).
\end{aligned}
\tag{47}
$$

In summary the ground state energy of the spin chain with non-diagonal boundary fields is

$$
E = E_0 + 4e(x_0) + 2\mathrm{i}\left[e' * \log A\big|_{\mathrm{i}} - e' * \log\overline{A}\big|_{-\mathrm{i}}\right].
\tag{48}
$$

Note that $E - E_0 = \mathcal{O}(1/N)$, because $e(x_0) \simeq \frac{\pi}{N}$ and the function $\log A(x) + \log\overline{A}(x)$ takes non-negligeable values only for arguments $|x| > x_0$. The calculation of the $1/N$ corrections involves the dilog-trick and quantitative calculations for the functions $A$, $\overline{A}$, $C$ in the scaling limit $N \to \infty$. These results will be communicated in separate publications.

# 5  Alternative contours

The main application of NLIEs is the study of finite size corrections for large system sizes. To this end, the above introduced functions $A$, $\overline{A}$, $C$ on the chosen contours are the means of choice. For finite system size the iterative numerical solution scheme suffers from certain instabilities as described in Sect. 4. We found – literally – ways to improve the stability of the iterative procedure: we consider the introduced functions on different integration contours. Instead of specializing the meromorphic function $B(z)$ on the contour $\mathbb{R} - \mathrm{i}$ we specialize it on a contour $\mathbb{R} - \delta\mathrm{i}$ lying above the two extremal roots $x_{2,r/l}$, cf. Fig. 1a, and below the other roots that are close to the real axis.

The functions $A(x) = B(x - \mathrm{i})$ and $A^{(\delta)}(x) = B(x - \delta\mathrm{i})$ thus defined can be analytically continued into each other, but have rather different properties: $A^{(\delta)}(x)$ does not show any winding, both asymptotics of $\operatorname{Im} \log A^{(\delta)}(x)$ are zero, see Fig. 4. This is because $q_2(x_{2,r/l}) = B(x_{2,r/l}) = 0$. Similarly, the zeroes of $q_1(z) = q_2(-z)$ and $\overline{B}(z)$ are $x_{1,r/l} = x_{2,r/l}^*$. This leads further to the following relations of convolution integrals of a function $k(x)$ with $\log A$ and $A^{(\delta)}$

$$(k * \log A)(x) = \left(k * \log A^{(\delta)}\right)(x - \mathrm{i} + \delta\mathrm{i}) + \mathrm{i}K(x - x_{2,l} - \mathrm{i}) + \mathrm{i}K(x - x_{2,r} - \mathrm{i}) \tag{49}$$

$$(k * \log \overline{A})(x) = \left(k * \log \overline{A}^{(\delta)}\right)(x + \mathrm{i} - \delta\mathrm{i}) - \mathrm{i}K(x - x_{1,l} + \mathrm{i}) - \mathrm{i}K(x - x_{1,r} + \mathrm{i}), \tag{50}$$

where $K(x)$ is the integral of $k(x)$, i.e. $K' = k$, and the additional terms result from residue contributions. The above relations are most tranparently checked in their differentiated form.

The integral equations for the functions $\log a^{(\delta)}(x)$, $\log \overline{a}^{(\delta)}(x)$, $\log c(x)$ are

$$\begin{aligned}
\log a^{(\delta)}(x) =& (2N + 1) \log \tanh\left(\frac{\pi}{4}(x + (1 - \delta)\mathrm{i})\right) + \frac{\pi}{2}\mathrm{i} + \log a_\infty - \mathrm{i}\alpha(x - \mathrm{i}, 1) \\
&+ \mathrm{i}\alpha(x - \delta\mathrm{i}, p_1) + \mathrm{i}\alpha(x - \delta\mathrm{i}, p_2) - \mathrm{i}\alpha(x - x_0 - \delta\mathrm{i}, 1) - \mathrm{i}\alpha(x + x_0 - \delta\mathrm{i}, 1) \\
&+ \mathrm{i}\alpha(x - x_{1l} - \delta\mathrm{i}, 1) + \mathrm{i}\alpha(x - x_{1r} - \delta\mathrm{i}, 1) + \mathrm{i}\alpha(x - x_{2l} - \delta\mathrm{i}, 1) + \mathrm{i}\alpha(x - x_{2r} - \delta\mathrm{i}, 1) \\
&+ K_{11}^{(\delta)} * \log(A^{(\delta)}/A_\infty) + K_{12}^{(\delta)} * \log(\overline{A}^{(\delta)}/\overline{A}_\infty) + K_{13}^{(\delta)} * \log(C/C_\infty), 
\end{aligned} \tag{51}$$

$$\begin{aligned}
\log \overline{a}^{(\delta)}(x) =& (2N + 1) \log \tanh\left(\frac{\pi}{4}(x - (1 - \delta)\mathrm{i})\right) - \frac{\pi}{2}\mathrm{i} + \log \overline{a}_\infty + \mathrm{i}\alpha(x + \delta\mathrm{i}, 1) \\
&- \mathrm{i}\alpha(x + \delta\mathrm{i}, p_1) - \mathrm{i}\alpha(x + \delta\mathrm{i}, p_2) + \mathrm{i}\alpha(x - x_0 + \delta\mathrm{i}, 1) + \mathrm{i}\alpha(x + x_0 + \delta\mathrm{i}, 1) \\
&- \mathrm{i}\alpha(x - x_{1l} + \delta\mathrm{i}, 1) - \mathrm{i}\alpha(x - x_{1r} + \delta\mathrm{i}, 1) - \mathrm{i}\alpha(x - x_{2l} + \delta\mathrm{i}, 1) - \mathrm{i}\alpha(x - x_{2r} + \delta\mathrm{i}, 1) \\
&+ K_{21}^{(\delta)} * \log(A^{(\delta)}/A_\infty) + K_{22}^{(\delta)} * \log(\overline{A}^{(\delta)}/\overline{A}_\infty) + K_{23}^{(\delta)} * \log(C/C_\infty),
\end{aligned} \tag{52}$$

$$\log c(x) = \log\left[c_\infty \frac{x^2\left(x^2 - x_0^2\right)}{(x - x_{1l})(x - x_{1r})(x - x_{2l})(x - x_{2r})}\right] + K_{31}^{(\delta)} * \log A^{(\delta)}/A_\infty) + K_{32}^{(\delta)} * \log(\overline{A}^{(\delta)}/\overline{A}_\infty).$$

Here the kernel matrix is

$$K^{(\delta)}(x) = \begin{bmatrix} \kappa(x, 1) & -\kappa(x - 2\delta\mathrm{i}, 1) & -\mathrm{i}/(x - \delta\mathrm{i}) \\ -\kappa(x + 2\delta\mathrm{i}, 1) & \kappa(x, 1) & \mathrm{i}/(x + \delta\mathrm{i}) \\ \mathrm{i}/(x + \delta\mathrm{i}) & -\mathrm{i}/(x - \delta\mathrm{i}) & 0 \end{bmatrix}. \tag{53}$$

Note that in this case, because of the absence of the winding phenomenon, no counter terms are necessary. Instead, we have explicit contributions to the source terms from functions depending on the locations of the extremal roots $x_{1,r/l}$ and $x_{2,r/l}$ (which are complex conjugate to each other). These roots are to be found from solving $a^{(\delta)}(x_{2,r} + \delta\mathrm{i}) = -1$ by evaluating the right hand side of (51) below the real axis.

Note that the functions $a$ and $A$ may be denoted $a^{(1)}$ and $A^{(1)}$, but they do not satisfy the above NLIEs. For this it would be necessary that the imaginary part of $x_{2,r/l}$ be smaller than $-1$.

The energy formula (48) together with (50) takes the form

$$E = E_0 + 4e(x_0) - 2e(x_{1,r}) - 2e(x_{1,l}) - 2e(x_{2,r}) - 2e(x_{2,l}) + 2\mathrm{i}\left[e' * \log A^{(\delta)}\big|_{\delta\mathrm{i}} - e' * \log \overline{A}^{(\delta)}\big|_{-\delta\mathrm{i}}\right]. \tag{54}$$

Evaluated for systems with boundary parameters $p = -0.6$, $q = -0.3$, $\xi = 0.5$ and system sizes $N = 12$ and $N = 24$, this yields energy values -23.7997... and -45.2062... which lie close to the results of Lánczos calculations -23.8004... and -45.2068... . The deviations are of order $\mathcal{O}(10^{-4})$ where the Lánczos data come with 14 digits accuracy. We are rather satisfied with these results in view of the convergence issues discussed above. In addition, the long-range nature of the Cauchy kernel requires careful treatment if truely high accuracies are intended. As pointed out at the beginning of this section, our main goal is the analytic treatment of finite size corrections in the thermodynamic/scaling limit.

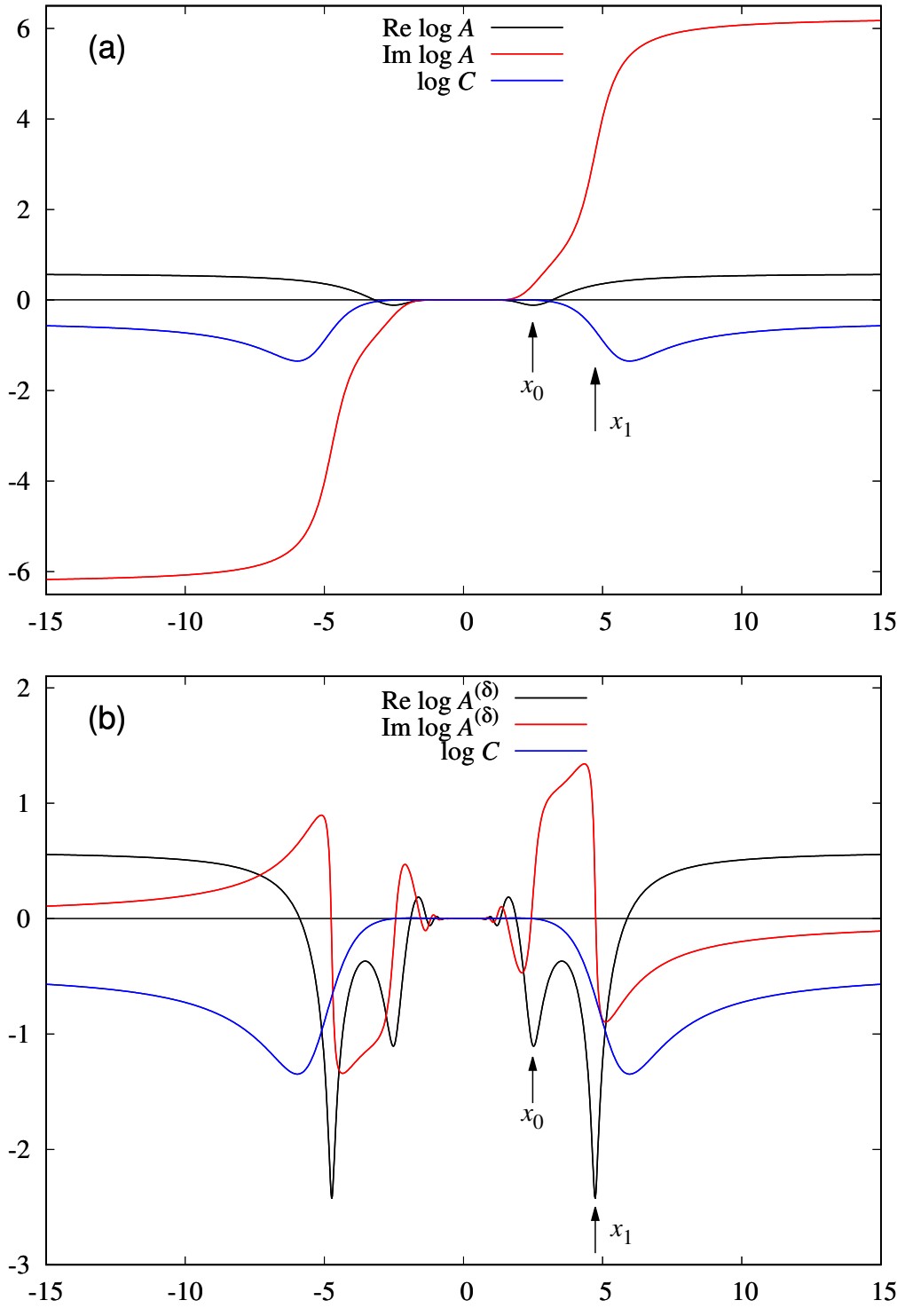

Figure 4: (a) Plots of the functions $\log A$ and $\log C$ for the system with boundary parameters $p = -0.6$, $q = -0.3$, $\xi = 0.5$ and system size $N = 12$. The arrows indicate the characteristic points $x_0$ (the zero of $\log C$) and $x_1$ (where the imaginary part of $\log A$ strongly increases from small values to $2\pi$). (b) Plots of the functions $\log A^{(\delta)}$ and $\log C$ (same data as in (a)) for the same parameters and $\delta = 0.25$. In this case no winding occurs: the imaginary part of $\log A^{(\delta)}$ tends to zero as the argument approaches $\pm\infty$. However, both the real and the imaginary parts of this function exhibit oscillations. The arrows mark the same locations as in panel (a). It now becomes clear that these points are associated with the real parts of two zeros of the function $A^{(\delta)}$: the real part of its logarithm assumes large negative values. For the chosen boundary parameters we have $x_0 = 2.4868...$ and $x_{2,r/l} = \pm 4.7322... - i \cdot 0.3365...$.

# 6 Conclusion

The XXX spin chain with non-diagonal boundary fields stands as a prominent yet challenging example of quantum integrable systems with non-trivial boundaries [8–10, 12–15, 17–20, 22–25]. Despite its relevance for the theory of integrable systems the exact solution has remained a longstanding problem. A great part of a satisfactory solution is realized by the derivation of inhomogeneous T-Q relations and the corresponding Bethe equations, discovered by the off-diagonal Bethe Ansatz (ODBA) [13, 14] and further understood and extended in other work [15, 17–20, 22–24]. The analytic treatment of these equations has so far been limited to the thermodynamic limit [30], without retaining terms that characterize finite-size corrections. While bulk and surface energies, as well as certain excitations, were analyzed in [25, 30], finite-size effects remained elusive. In this work, we successfully construct nonlinear integral equations (NLIEs) to tackle this challenge.

A key advancement in our work is the introduction of the function $c(x)$, which accounts for the inhomogeneous term in the T-Q relation. This complements the classical functions $a(x)$ and $\bar{a}(x)$ and allows for the complete description of the system. When the boundary fields are taken parallel the NLIEs simplify significantly, reducing to two coupled equations without long-range kernel terms. These simplified equations are computationally efficient to solve. However, for non-parallel boundary fields, i.e. non-zero values of the parameter $\xi$ the situation is different. First of all, we observe the winding phenomenon, the functions $\log A(x)$ and $\log \bar{A}(x)$ show sudden changes by $2\pi i$ at some characteristic scale $x_1$ of the argument. At a rather different value $x_0$ the function $\log C(x)$ turns zero. Second, we realized that the two scales $x_0$ and $x_1$ are independent. We succeeded in obtaining explicit numerical results for large and small values of the ratio $x_1/x_0$ which are taken for small and large system size $N$.

This study represents the first step in our broader project to explore the conformal properties of the Heisenberg spin chain through the lens of NLIEs. The large $N$ results allow for future analytical study of the finite size properties of the system with non-parallel boundary fields. The next step will be the derivation of a suitable scaling limit of the NLIEs. Preliminary studies have yielded a simplified kernel matrix still with the same long-range terms, but all regular terms being simplified to delta-functions. The full understanding of this and the combined analytical-numerical investigations require additional work that will be published elsewhere.

By starting with the isotropic XXX model, we have developed a framework that paves the way for generalizations to the XXZ spin chain with arbitrary open boundary conditions. Our approach is designed to address the challenges posed by the loss of $U(1)$ symmetry in these systems and to provide a robust method for analyzing finite-size corrections and related properties.

Ultimately, we aim to contribute not only to the theoretical understanding of integrable models but also to their application in broader contexts, such as statistical mechanics and condensed matter physics, where boundary effects play a critical role.

# Acknowledgment

X.Z. acknowledges financial support from the National Natural Science Foundation of China (No. 12204519) and from the Alexander von Humboldt Foundation. A.K., D.W. and H.F. acknowledge financial support by Deutsche Forschungsgemeinschaft through FOR 2316. A.K. acknowledges hospitality by the Innovation Academy for Precision Measurement Science and Technology, Wuhan, and funding by the Chinese Academy of Sciences President's International Fellowship Initiative, Grant No. 2024PVA0036.

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
