# Peer review of "Non-linear integral equations for the XXX spin-1/2 quantum chain with non-diagonal boundary fields"

_SciPost Physics_

## Round 1 · Author Response

Dear Editors,

We hereby resubmit our manuscript to SciPost. The scientific results and conclusions remain identical to those in the original submission. In response to the referees’ valuable feedback, we have substantially improved the clarity and presentation of the work, added several explanatory comments throughout the manuscript, and included a new two-page Section 5 containing additional results that further strengthen our main findings.

We believe that the revised version addresses all points raised by the referees and hope that it now meets the standards for publication in SciPost.

Sincerely,
Holger Frahm, Andreas Kl\"umper, Dennis Wagner, Xin Zhang

---

## Round 1 · List of Changes

Warnings issued while processing user-supplied markup:

  • Inconsistency: plain/Markdown and reStructuredText syntaxes are mixed. Markdown will be used.
    Add "#coerce:reST" or "#coerce:plain" as the first line of your text to force reStructuredText or no markup.
    You may also contact the helpdesk if the formatting is incorrect and you are unable to edit your text.

We thank Referee 1 for the critical and detailed questions raised about our work. We appreciate these comments, as they have given us the opportunity to further improve the manuscript.

We thank Referee 2 for viewing our manuscript in its proper scientific and historical context and for the strong support expressed regarding its suitability for publication in SciPost.

We thank Referee 3 for the helpful questions and constructive suggestions. We hope that our revisions address these points satisfactorily.

All changes and additions to the manuscript are highlighted in blue in the newly submitted version. We would like to emphasize that a new Section 5, comprising about two pages, has been added. This new section, as well as all other new or revised passages responding to the referees’ comments, is typeset in blue.

Below we answer the questions and comments of the referees and point to the relevant positions of amendments in the manuscript.

———————————– Report 1 ————————————

In this paper, the authors present a new analysis to the isotropic Heisenberg chain with arbitrary boundary fields, explicitly retaining terms that depend on the angle between the boundary field. I believe it deserves to be published in Scipost after the following comments have been appropriately addressed:

  1. Reviewer: Above Eq(5)-(7), the authors state: ” In analogy to [26, 27], we find that the following functions are useful” It would be helpful to further clarify the physical motivation for adopting these particular functional forms. For example, do they offer specific advantages in numerical implementations? Moreover, are there alternative formulations that might also be viable?

Answer: Yes, in the manuscript above the definition of the functions we write "In analogy to [26, 27], we find that the following functions are useful". And the usefulness lies in the factorization of the functions in terms of q_1, q_2, D, \bar D, \Lambda shown in (10). And interestingly those functions can be eliminated and integral expressions for b, \bar b, c in terms of B, \bar B, C result. That this is possible is not clear when just looking at (5)-(7). There is no greater theoretical background for this.

In sections 3 and 4 the NLIEs are written down for the functions log b(z), log \bar b(z), log c(z) etc. specified to (evaluated on) the contours: real axis -i, real axis + i, real axis, where the functions are called log a(x), log \bar a(x), log c(x) etc. We added a new section 5 where different contours are used that allow to avoid the winding phenomenon at the expense of having to deal with functions with oscillations, fortunately with very small amplitudes. We believe that this section adds some insight to the understanding of the "behaviour of the functions".

  1. Reviewer: Below Eq(10), the authors state: ”We want to ’solve’ these functional equations of multiplicative type with constant shifts in the arguments. The first step is the application of the logarithm, turning the product form into additive form. In the second step, we apply the Fourier transform, which turns functions with shifts in the arguments into transforms with simple factors, eventually leading to a set of linear equations. Care has to be taken that the Fourier transform exists and that the region of convergence of the inverse transform–the Fourier representation–is wide enough for our purposes.” It would be helpful if the authors could elaborate on the role and necessity of the Fourier transform in this formulation. In particular, what are the specific advantages it offers in the numerical iteration?

Answer: We added on page 4 at the end of the cited paragraph:

"At the end of these procedures, the non-linear integral equations emerge. The strategies for solving these equations are largely numerical and are described in Sect. 4."

  1. Reviewer: In Figure 1, the authors state: ”Depiction of Bethe roots and zeros of the eigenvalue function for the ground-state and parameters {p, q, ξ} = {−0.6, −0.3, 1.2}” Could the authors clarify whether the zero root distribution presented in this work is valid across the full range of boundary parameter values? If not, it would be helpful to provide a comprehensive list or classification of the distinct root distributions associated with different boundary parameters. Additionally, above the Eq.(11), the authors state ”In case of the largest eigenvalue...”. A clarification is needed as to whether this particular distribution of zero roots corresponds to the ground state or to the state associated with the largest eigenvalue.

Answer: We added to the caption of Fig.1:

"This choice is representative of the situation in which both p and q are negative."

We have investigated other combinations of the parameters p and q, but we prefer to reserve these results for a future publication, as providing further details here would risk sidetracking the discussion and causing confusion.

We are sorry about the confusion about the "largest" and "lowest" eigenvalues. We added after (8):

"Numerical calculations for the largest eigenvalue of the transfer matrix--corresponding to the lowest eigenvalue of the Hamiltonian--show that $N$ zeros of $\Lambda(x)$ have imaginary parts greater than $+1$..."

  1. Reviewer: Above the Eq. (18), the authors state: ”In any case, for large system size, the bulk of the zeros of D(x), respectively D̄(x), is close to Im(x) = +2, respectively Im(x) = −2.” It would be helpful if the authors could clarify, from a numerical perspective, the minimal system size required for the stated distribution of zeros of D(x) to manifest. Furthermore, does this imply that equations (35)-(37) are only valid in the large-system-size limit?

Answer: We like to leave the statement given in the paper "This is also the case for all system sizes if p1 , p2 > 1 and certain cases of p1 > 1 > p2 > 0 which is what we consider in this paper".

It may also be that the stated properties about the zeros of D(x) hold for all system sizes N for all cases of p1 > 1 > p2 > 0. For 0<p1, p2<1 there is an upper bound on N for the stated property of the function D(x). We certainly exclude 0<p1, p2<1 from our analysis.

  1. Reviewer: Above the Eq. (23), the authors state: ”We will factor out of a(x) and ā(x) an explicit function yielding the zero of high order. It will appear that, in some of the formulas below, the shift x → x ± i should be understood as x → x ± (1 − ε)i with small positive ε.” Could the authors provide a detailed explanation of how to factor out of a(x) and ā(x) an explicit function yielding the zero of high order? Additionally, is the introduction of the small positive parameter ε intended to numerically regularize potential divergences? In equations (35)(37), this small parameter appears only in K12 and K21 . Could the authors clarify why the regularization is required exclusively for these two terms? Finally, with regard to Figure 2, what specific value of ε was used during the numerical iteration? It would be helpful if the authors could provide comparative plots for different values of ε to numerically demonstrate that the results are indeed insensitive to its precise value.

Answer: The comment above (23) was premature and therefore potentially misleading, so we have revised it. The remark concerning the factorization of a high power of a \tanh function has been removed entirely, as it is not essential. The discussion of the role of the \epsilon parameter in numerical calculations has been relocated to the end of Section 3. In brief: In Fourier space we are allowed to set \epsilon = 0 in kernel functions where poles are sufficiently far from the integration contour. And for kernel elements with poles at a distance \epsilon from the integration contour, the use of the fast Fourier transform allows us to set \epsilon = 0 also there.

  1. Reviewer: Above the Eq. (38) in Section 3, the authors state: ”where the convolutions in (37) are done with log A and log Ā evaluated on symmetric intervals [−R, +R] with R → ∞.” Could the authors clarify how the expression log(∞) should be interpreted in this context?

Answer: We have slightly rewritten this form of NLIEs also in view of the new Sect. 5 where a similar form of NLIEs appears that we actually use.

  1. Reviewer: About the Eq. (39) in Section 4, the authors state: ”After formulating the NLIEs for the derivatives (log a(x))′ , . . . , (log A(x))′ , . . ., the NLIEs are integrated with respect to the argument x, and the integration constants are fixed by considering the limit x → ∞. The result is...”. It would be helpful if the authors could elaborate on the derivation of equation (39) in more detail.

Answer: The missing information now appears after eq. (42).

  1. Reviewer: Could the authors provide more details on how equation (43) is obtained or why you choose these numbers? Furthermore, in the numerical implementation, what specific value is chosen for the parameter δ, and does this choice affect the outcome of the iterative solution? It would be helpful to clarify whether the results are robust with respect to variations in δ.

Answer: The values of the real and imaginary parts of the parameters appearing in (43) have indeed been chosen as indicated in the text. The choice of \tilde x_1 is obvious, it is the location of the transition in the imaginary part of \log A. The \delta parameter (now \tilde\delta) could be chosen to model the steepness of the transition, but we did not care about that as the precise values drop out anyway.

The new Sect. 5 of the current manuscript allows to relate the "optimal parameters" appearing in (43) to the locations of the extremal roots mentioned in Fig.1.

  1. Reviewer: Below Eq(43) in section 4, the authors state: ”By use of qualitatively similar initial data for the iterative treatment of the NLIEs, we found convergence for much larger system sizes, like that shown in Fig. 2 for N = 10^3 . However, increasing the system size further will ultimately, and independently of the chosen boundary parameters, lead to the loss of convergence.” It would be helpful if the authors could elaborate on how the initial data are distributed in the iterative approach. Additionally, is it feasible to reproduce similar structures to those in Fig. 2 for smaller system sizes such as N = 6, 8, 10?

Answer: We now explain the choice of initial data on page 10 of the manuscript. An analogue to Fig.2 for smaller system size (N=12) appears as Fig.4(a) in the new section.

  1. Reviewer: Above the Eq. (44) in Section 4, the authors state: ”Under this condition, the above-described winding happens so fast that the convolution integrals produce contributions that cancel the leading (2N + 1) log tanh (π/4 x) . On the basis of this reasoning, we found solutions of the NLIEs with much larger system sizes, like those illustrated in Fig. 3.” Is Figure 3 obtained by performing the numerical iteration based on equations (35)-(37) after omitting the leading term (2N + 1) log tanh (π/4 x) , or does it result from using equation (39), after omitting the leading term (2N + 1) log tanh (π/4 x) ? A clarification on this point would be appreciated.

Answer: We never used (35)-(37) in numerical calculations. We used (39) and after locating the transition point we found out that the extremal roots responsible for the transition in the imaginary part of \log A approach the integration contour with a vanishingly small distance (~10^-10). Hence the numerical data show a jump from 0 to 2\pi at x_1.

  1. Reviewer: Is there any difference in the numerical results obtained by iterating equations (35)-(37) as opposed to equation (39)? Should the outcomes be identical in principle? If so, it would be helpful if the authors could provide a comparative plot to demonstrate the agreement between the two approaches.

Answer: Like said before, we never used (35)-(37) for numerics.

  1. Reviewer: In conclusion, the authors state: ”The large N results allow for future analytical study of the finite size properties of the system with non-parallel boundary fields. The next step will be the derivation of a suitable scaling limit of the NLIEs.” This part of the analysis may be diffcult to follow, but in this context, it seems insuffcient to limit the numerical results to the distribution of the A function. A direct comparison of the ground state energy with independent DMRG data at fixed, finite system sizes but not in the thermodynamic limit would be essential to substantiate the correctness of the result. Could the authors comment on this and possibly include such a comparison?

Answer: We wrote the new Sect. 5 to present a form of the NLIEs that is more suitable for calculations for small values of N. There we compare the ground-state energy data obtained from the NLIEs for two cases N=12 and 24 with Lánczos data. The result confirms that the presented NLIEs are correct. In the current form of the numerical implementation of our NLIEs the accuracy is not yet very satisfying. However, the main goal of applications of the NLIEs is the scaling limit.

———————————– Report 2 ———————————–

In this article the authors develop and apply the nonlinear integral equation (NLIE) approach to the calculation of physical properties of the spin-1/2 XXX Heisenberg chain with non-diagonal boundary fields. Although this model is known to be Yang-Baxter integrable for the most general non-diagonal boundary terms, calculating physical properties of interest for this model has remained a challenge due to the breaking of the U(1) symmetry by the boundary fields. Progess over the years by various approaches, most notably the off-diagonal Bethe Ansatz method, has been applied by the authors to develop a set of NLIEs capturing the spectrum of the model with non-diagonal boundary fields. Over several decades the NLIE approach has been applied previously, along with other methods, to successfully derive physical properties for the special case of diagonal boundary fields, including via finite-size effects.

It is seen that a key feature of the NLIE approach developed for the model with non-diagonal boundary fields is the necessity to introduce an additional third function c(x) . In their preliminary analysis of the NLIEs the authors reproduce the known results for the bulk and boundary contributions for the ground state energy.

In summary this article represents a significant step forward in the study of the spin-1/2 XXX Heisenberg chain with non-diagonal boundary fields, by extending the NLIE approach to calculate bulk and boundary properties and lay the groundwork for future calculations of finite-size corrections and related properties. This progress via the NLIE approach is also made with an eye towards applyication to other models with U(1) symmetry breaking boundary terms, the most immediate being the XXZ chain. I have no hesitation in recommending publication of this article.

Answer: We thank Referee 2 once again for viewing our manuscript in its proper scientific and historical context and for the strong support expressed regarding its suitability for publication in SciPost.

———————————– Report 3 ———————————–

Report on "Non-linear integral equations for the XXX spin-1/2 quantum chain with non-diagonal boundary fields" by H. Frahm, A. Kluemper, D. Wagner, and X. Zhang

The manuscript is devoted to the derivation and analysis of the nonlinear integral equations, which solution describes the behavior of the isotropic (antiferromagnetic) spin-1/2 Heisenberg chain with fixed boundary conditions. The latter is achieved by the application of local boundary magnetic fields, one of which is coupled to two projections of the edge spin. The nonlinear equations are derived using so called inhomogeneous T-Q relations.

On the one hand, the topic of the manuscript is interesting, and the derivation of the nonlinear integral equations and their analysis is interesting and deserves publication.

On the other hand, the presentation of the results, to my mind, needs some changes, several corrections and additions are necessary:

(i) It is not clear from the text that the treatment addresses only features of the ground state. Also, I assume (though it is not written directly) that the number of sites of the chain is even. Those issues can be added to the manuscript.

Answers: The evenness of the number of sites is spelled out after (1). Our analysis indeed targets the ground-state which is pointed out many times in the manuscript.

(ii) The main achievement of the manuscript is the derivation of the nonlinear integral equations for the considered case. The authors state that unlike the set for periodic boundary conditions, the derived set contains three (not two) functions, which depend on spectral parameter and the values of boundary fields. In fact, I do not understand that comparison. The known solution of two nonlinear integral equations for periodic boundary conditions (for functions a and \bar a), describes the case of NONZERO temperature, see below, and it is namely the advantage of the so-called quantum transfer matrix method. Here the authors consider the T=0 case only. To my mind, the comparison is unclear. It seems more reasonable to compare obtained description with the ones obtained by other methods, see below.

Answer: The method of NLIE can be and has been applied to the QTM which contains the finite temperature properties of a quantum system. But also and in fact earlier, the NLIE have been applied to the row-to-row transfer matrix that contains all information on the eigenstates of the quantum system, notably those of the ground-state. Many of the mathematical techniques used for T=0 and finite N on one hand and for T>0 and N=infinity on the other hand are very similar, the physics that is described is rather different.

(iii) The abstract contains the invalid statement. It is written "for the non U(1) symmetric case, the equations involve a novel third function c(x)". First, for the non U(1) symmetric case one can describe thermodynamics of the XYZ chain (clearly non U(1) symmetric case) using only two functions, a and \bar a, published many years ago by the second author, A. Kluemper, Z. Phys. B 91, 507 (1993). Perhaps the authors meant non U(1) symmetry caused by boundary fields. Second, one can see from the derived set of nonlinear equations that for the absence of the "non-diagonal" term in the Hamiltonian, \xi =0, one still has three nonlinear equations, including the one for c, at least some procedure to get two equations in that case has to be applied and explained.

Answers: Note, the status of the XYZ chain, i.e. the topics "non U(1) symmetric bulk interactions and periodic boundary", and "U(1) symmetric bulk interaction and U(1) symmetry breaking boundary fields" are rather different.

In the abstract we write about a spin chain with SU(2) invariant bulk interaction. The "U (1) symmetric case" and the "non U (1) symmetric case" are always meant with respect to the boundary fields. For clarity we replaced:

For the non U (1) symmetric case, the equations involve a novel third function c(x)... -> In the case considered here the $U(1)$ symmetry is broken by the non-diagonal boundary fields and the equations involve a novel third function $c(x)$, which captures the inhomogeneous contributions to the $T$-$Q$ equation in the ODBA.

Note that the spin 1/2 XYZ chain with even number of sites and periodic boundary condition satisfies a set of two NLIEs. And this is unrelated and does not give a hint in the analysis of the spin 1/2 XXX chain with boundary fields breaking U(1) symmetry.

And yes, for \xi =0 the third equation "disappears": Whatever the integral contributions to log(c) in (39) are, the driving term in (42) containing c_\infty=0 makes c(x) identical to 0. Hence log(C/C_infty) = 0 and does not contribute to the two functions a(x) and \bar a(x).

(iv) In fact, the title implies the special role of the "non-diagonal boundary fields". However, it is not clear from the analysis that one needs namely those terms. Let us look at the main results of the manuscript, Eqs. (35)-(38), the nonlinear equations, and Eqs.(44)-(46), the ground state energy. If one puts \xi =0 to those equations, no qualitative changes appear, see also the remark (iii). Where is then the mentioned special role of the non-diagonal boundary terms?

Answer: Well, the case \xi=0 is very special. For non-zero \xi we have three coupled functions resp. equations and the winding phenomenon shown in Fig.2. For \xi=0 there are only two functions to be calculated and they do not show the winding as the parameter x_1 sits at infinity. Even if we put \xi=0, the Eqs.(44)-(46) do not reduce to those of the periodic case. And actually, we are not interested in \xi=0, it is \xi>0 what is the topic here.

(v) Unfortunately, there is not enough comparison with known results for boundary fields for spin-1/2 chains. For example, the case with \xi =0 has to coincide with the well-known results of P.A. de Sa and A.M. Tsvelik, Phys. Rev. B 52, 3067 (1995), A. Kapustin and S. Skorik, J. Phys. A 29, 1629 (1996). Non-diagonal boundary fields were considered, e.g, for the general XYZ chain W.L. Yang and Y.Z. Zhang, Nucl. Phys. B 744, 312 (2006), and reviewed in Ref. [14], for the XY chain A.A. Zvyagin, Phys. Rev. Lett. 110, 217207 (2013), and reviewed for the XX chain in A.A. Zvyagin, Finite Size Effects in Correlated Electron Systems: Exact Results, Imperial College Press, (2005). It is not clear from the text of the considered manuscript what are the special features of the obtained in the manuscript results comparing with the mentioned cases.

I support publication of the manuscript provided necessary corrections and additions made.

Answer: The physics and the mathematics of integrable quantum systems vary so much even when changing "innocent parameters" only a little. Therefore comparison with the suggested references is helpful only in a few cases (but thank you very much for letting us know the references):

P.A. de Sa and A.M. Tsvelik, Phys. Rev. B 52, 3067 (1995): concerns the thermodynamics of the open boundary XXZ chain, the T->0 limit can be taken, the boundary field dependent part of the ground-state energy can be obtained and agrees with our surface energy -- we cite this reference as [32] --

A. Kapustin and S. Skorik, J. Phys. A 29, 1629 (1996): among others, they calculate the surface energy for XXZ with diagonal boundary field, our results agree with their's in the XXX limit -- we cite this reference as [31] --

The paper W.L. Yang and Y.Z. Zhang, Nucl. Phys. B 744, 312 (2006) concerns an XYZ chain with boundary fields satisfying certain conditions allowing for the derivation of a homogeneous TQ-relation. This is not the case we study.

The papers on XY and XX chains are not comparable at all.

---

## Editorial Decision

accepted_in_target_journal